# Mitochondrial Metabolism in PDAC: From Better Knowledge to New Targeting Strategies

**DOI:** 10.3390/biomedicines8080270

**Published:** 2020-08-03

**Authors:** Gabriela Reyes-Castellanos, Rawand Masoud, Alice Carrier

**Affiliations:** Centre de Recherche en Cancérologie de Marseille (CRCM), Aix Marseille Université, CNRS, INSERM, Institut Paoli-Calmettes, F-13009 Marseille, France; gabriela.reyes-castellanos@inserm.fr (G.R.-C.); masoudrawand@gmail.com (R.M.)

**Keywords:** pancreatic ductal adenocarcinoma, cancer metabolism, mitochondria, mitochondrial metabolism, energetic metabolism, OXPHOS, metabolic heterogeneity, mitochondrial complex I, biguanides, therapeutic strategy

## Abstract

Cancer cells reprogram their metabolism to meet bioenergetics and biosynthetic demands. The first observation of metabolic reprogramming in cancer cells was made a century ago (“Warburg effect” or aerobic glycolysis), leading to the classical view that cancer metabolism relies on a glycolytic phenotype. There is now accumulating evidence that most cancers also rely on mitochondria to satisfy their metabolic needs. Indeed, the current view of cancer metabolism places mitochondria as key actors in all facets of cancer progression. Importantly, mitochondrial metabolism has become a very promising target in cancer therapy, including for refractory cancers such as Pancreatic Ductal AdenoCarcinoma (PDAC). In particular, mitochondrial oxidative phosphorylation (OXPHOS) is an important target in cancer therapy. Other therapeutic strategies include the targeting of glutamine and fatty acids metabolism, as well as the inhibition of the TriCarboxylic Acid (TCA) cycle intermediates. A better knowledge of how pancreatic cancer cells regulate mitochondrial metabolism will allow the identification of metabolic vulnerabilities and thus novel and more efficient therapeutic options for the benefit of each patient.

## 1. Background

Over the past decade, cancer researchers have turned their attention to the cellular organelle mitochondrion. Several studies have reported the critical role of mitochondrial metabolism in all facets of cancer: Development, progression and invasion. Moreover, mitochondria have been involved in resistance to chemotherapy. Targeting mitochondrial metabolism is thus an exciting area of research that has shown great promise in the treatment of various types of cancer.

Specifically, pancreatic cancer is a disease that could greatly benefit from mitochondrial metabolism targeting [1,2,3,4,5]. The most frequent pancreatic cancer is Pancreatic Ductal AdenoCarcinoma (PDAC), the fifth leading cause of cancer death in the United States by 2017 [6] and the 13th worldwide by 2018 [7]. With a five-year overall survival that has increased very slightly over decades (currently 8%), PDAC remains a poor-outcome disease with rising incidence in developed countries [8,9].

A major concern is that PDAC death rates have been rising over the past decade, unlike other cancers (e.g., lung, breast, prostate and colorectal) [6]. Moreover, PDAC is predicted to become the second cause of cancer-related deaths in the United States by 2030, surpassing breast and colorectal cancers [6,10,11]. There is thus an imperative need to invest massively to develop effective therapeutic strategies against this incurable disease.

The aggressive behavior of PDAC makes it difficult to treat, in addition to the fact that most patients are diagnosed in an advanced or even metastatic state [8]. In these patients, a combination of cytotoxic therapies as Gemcitabine plus Nab-Paclitaxel or FOLFIRINOX provides only a mild increase in survival (in the rank of weeks to months) [12,13]. The outcome could be better with a therapeutic strategy that would include inhibitors of mitochondrial metabolism for the patients whose tumor depend on mitochondria for aggressiveness and resistance to therapy.

Among inhibitors of mitochondrial metabolism, those that target the respiratory complex I (i.e., Metformin, Phenformin) are by far the most frequently cited. Furthermore, analogues or modified-Metformin compounds have been developed to better amplify the therapeutic activity of Metformin, as well as the combination with glycolysis inhibitors.

Nevertheless, due to the multifunctionality of mitochondria, there is a vast opportunity to explore further potential therapeutic options. In particular, the metabolism of nutrients such as glutamine and other amino acids, fatty acids, intermediates of the TriCarboxylic Acid (TCA) cycle, autophagy and the generation of Reactive Oxygen Species (ROS) are of main interest in the pancreatic cancer field.

In this review, we will first present the classical view of cancer metabolism over many decades, as well as an extensive analysis of several important aspects leading to the current view of cancer metabolism. The latter is a comprehensive and integral approach to better understand cancer metabolism and metabolic dependencies, with a special focus on the specifics of pancreatic cancer. We will then present current strategies targeting mitochondrial metabolic pathways that are vulnerabilities. A better knowledge of how pancreatic cancer cells regulate their metabolism, in particular mitochondrial metabolism, will allow the design of novel and more efficient therapeutic options for the benefit of each patient.

## 2. Cancer Metabolism: A New Perspective

### 2.1. One Hundred Years of Cancer Metabolic Reprogramming

Metabolism is the ensemble of biochemical reactions necessary for organism homeostasis and survival. It is now well established that metabolism in cancer cells is adapted to their high needs in energy and macromolecules supporting their aberrant proliferation. Even though the first observation of a metabolic alteration in cancer cells was made one century ago (“Warburg effect” or aerobic glycolysis) [14], cancer metabolism became only recently a very active research field, as it provides great opportunities for cancer diagnosis and treatment.

The term of metabolic reprogramming (also called metabolic rewiring, deregulation, changes, alterations) started to be used at the beginning of the past decade. Nowadays, metabolic reprogramming is an established hallmark of cancer that describes the ability of neoplastic cells to adjust energy metabolism and synthesis of macromolecules to sustain cell growth and division [15]. Importantly, this capacity of metabolic rewiring is common to almost all cancers and it exists during all stages of cancer progression, so it is practically possible to consider cancer metabolism as a synonym of metabolic reprogramming.

The metabolic properties of tumoral cells are altered with regards to their normal counterpart. Normal cells respond to growth factor stimuli by the activation of canonical signaling pathways with the main outcome to support anabolism function. In contrast, cancer cells are able to fulfill their anabolic needs with minimal dependence on extrinsic stimuli by growth factors [16], through deregulation of pathways related to energy production and biosynthesis (Figure 1). Among these pathways, glycolysis and glutaminolysis are by far the most investigated. Accordingly, the most common signaling molecules involved in metabolic pathways are the phosphatidylinositol 3-kinase (PI3K), protein kinase B (known as AKT), adenosine monophosphate-activated protein kinase (AMPK) and the mammalian target of rapamycin (mTOR), as well as the oncogenic proteins KRAS and MYC and the tumor suppressor p53 [16,17].

The classical cancer energetic metabolism view describes the “Warburg effect” or aerobic glycolysis, characterized by high glycolytic rates observed in cancer cells even in the presence of oxygen, with a consequent high amount of lactate production (Figure 1). This phenomenon stands for a reliance of cancer cells on a glycolytic phenotype, instead of an oxidative phenotype (given by mitochondria). However, since glycolysis-produced ATP is far less efficient than oxidative phosphorylation (OXPHOS) in mitochondria, there are different theories regarding the possible advantages that this glycolytic phenotype provides to cancer cells [17]. The most accepted explanation for this preferred pathway is that glycolytic intermediates support macromolecular synthesis, such as that of nucleotides, lipids and some amino acids, as well as reduced equivalents in the form of NADPH [18,19]. Other theory is that even if the yield of ATP production by aerobic glycolysis is less efficient (18 times less), ATP is synthetized 100 times faster than OXPHOS, thus rapidly supplying energy demands [19,20]. Other proposed benefits of the Warburg effect are that it promotes invasion by enhancing tissue disruption and immune cell evasion due to acidosis in the tumor microenvironment. Finally, aerobic glycolysis could confer direct signaling functions to cancer cells, mainly the regulation of Reactive Oxygen Species (ROS) and the mediation of chromatin state [19].

Nonetheless, the current view of cancer energetic metabolism points to different several aspects, leading to a more integral and comprehensive understanding of this concept. First, this new perspective encompasses the shared roles of glycolytic and oxidative metabolism in cancer (Figure 1) and the factors that influence the preference of one pathway over another by cancer cells. Second, there is a vast list of molecules besides glucose that fuel cancer metabolism, as well as different means to obtain these molecules. Finally, this current perspective highlights the complexity of tumor metabolism in three key features: (a) Heterogeneity, (b) flexibility and (c) adaptability during cancer progression. More importantly for the scope of this review, mitochondrial functions are crucial in all the cancer metabolism aspects stated above.

#### 2.1.1. Glycolytic, OXPHOS or Hybrid Phenotype?

In addition to the extensively described glycolytic phenotype, there is now significant evidence that most cancers also rely on mitochondria to satisfy their metabolic needs [21]. In fact, mitochondria are quite active and yield most of the ATP in cancer cells, as in their normal counterparts [22]. Hence, it is established that both glycolysis and mitochondrial metabolism (through OXPHOS) are able to maintain anabolic growth (biosynthesis) and catabolism (bioenergetics), although one pathway usually dominates within a given cell.

Importantly, tumors are not metabolically homogeneous and the glycolytic or oxidative phenotype can depend on nutrient availability and the microenvironment. For instance, Shiratori et al. [23] demonstrated that the suppression of glycolysis in different cancer cells including pancreatic (Panc-1 cell line), induces a dramatic switch towards the OXPHOS profile. This switch is characterized by an increase in enzymes involved in TCA cycle and OXPHOS, as well as an increase in mitochondrial dynamics and mitochondrial membrane potential. Moreover, the glycolytic phenotype is associated with a cellular adaptation to hypoxic conditions, driven by activation of the hypoxia-inducible factor 1 (HIF-1), activation of oncoproteins (e.g., KRAS) and loss of tumor suppressor functions (e.g., p53). Accordingly, in a PDAC mouse model, the oncoprotein KRAS stimulates glucose uptake and channeling of glucose intermediates into the hexosamine biosynthesis and pentose phosphate pathways (PPP), thus promoting the glycolytic phenotype [24].

Using a systems biology approach, Jia et al. [25] determined that cancer cells can acquire three stable metabolic phenotypes: Glycolytic, characterized by high HIF-1 and low AMPK; OXPHOS, distinguished by low HIF-1 and high AMPK; and hybrid, with both high HIF-1 and AMPK. Interestingly, a hybrid glycolytic/OXPHOS phenotype can provide several benefits over cells using only one pathway, including the flexibility to use different nutrients and a more efficient energy production. Importantly, the hybrid phenotype can promote resistance to therapies and dissemination through metastasis.

In conclusion, cancer cells can exhibit distinct metabolic phenotypes driven by nutrient and oxygen availability, as well as key oncogenes, transcription factors and tumor suppressor genes.

#### 2.1.2. The Different Fuels Feeding Cancer

Besides glucose, other fuels contribute to core metabolic functions in cancer, and glutamine is one of them. Glutamine is a nonessential amino acid considered a major nutrient source for many cancer cells, and its uptake is significantly enhanced in cancer cells along with glucose. In particular, it has been demonstrated that pancreatic cancer cells are profoundly sensitive to glutamine deprivation [26]. However, preferential glutamine metabolism was not observed in KRAS-driven lung tumors, comparing with cells in culture, pointing out the influence of tissue environment on tumor metabolic phenotypes [27].

Glutamine is metabolized to glutamate via glutaminase and afterward to the TCA cycle intermediate α-ketoglutarate by dehydrogenase or transaminase enzymes. Indeed, glutamine is an important source of TCA cycle anaplerosis and along with glutamate it provides nitrogen for the production of other amino acids such as serine, alanine, aspartate, asparagine, proline and arginine [28,29]. Asparagine in particular has attracted the interest of researchers because it has been shown that this amino acid is crucial in sustaining cancer survival under glutamine depletion [30,31]. Of interest, the essential amino acid methionine has been an emerging focus in cancer metabolism, specially due to the antineoplastic effect of methionine-restricted diets [32]. Finally, other relevant fuels in cancer include the essential amino acids leucine, isoleucine and valine (branched chain amino acids, BCAA), mainly by providing precursors for protein and nucleotide synthesis [33].

In addition to glucose and amino acids, lipids constitute a relevant source for cellular membrane formation, second messengers and signaling molecules production and a substrate for energy generation and storage [34,35]. Moreover, cancer cells reactivate lipogenesis, making it a key metabolic footprint of almost all cancers and required for tumorigenesis, cancer progression and even cancer aggressive behavior [34]. Accordingly, de novo fatty acid synthesis (FAS) is a prominent hallmark of cancer [36] since fatty acids are necessary for the biosynthesis of most lipids [35]. However, lipid metabolism includes not only the process of lipid synthesis and storage, but also catabolism. Of great importance for this review, the catabolism of fatty acids occurs in mitochondria via the Fatty Acid Oxidation (FAO). FAO is a crucial pathway feeding the TCA cycle and is a source of both ATP and NADH. The role of FAO in cancer has remained obscure for a long time; however, over the last years, a growing body of evidence of the importance of FAO has accumulated in many types of cancer [29,36,37,38,39]. Finally, other reported sources that sustain cancer include the glucose sub products lactate and pyruvate, and also β-hydroxybutyrate and acetate [16,20,40].

The expanding list of cancer fuels provides a parallel increase in novel pathways required for their production, which can be targeted for cancer therapy. This is logical considering that cancer cells must compete for nutrients in a crowded tissue environment [40]. In consequence, cancer cells not only obtain nutrients by conventional pathways, but also rely on autophagy and scavenging of macromolecules [41,42,43]. Finally, which fuel is favored by a given cancer cell is influenced by both cell-intrinsic and extrinsic factors, such as gene mutations, nutrient and oxygen availability, as well as microenvironmental conditions [29].

There is thus broad evidence that targeting the manner cancer cells feed themselves and the pathways they are dependent on is a very relevant approach for cancer treatment. Importantly, the scope of this approach goes beyond the use of drugs and includes the regulation of nutrients availability through the diet. Ultimately, this is crucial for the treatment of such a devastating disease as pancreatic cancer.

#### 2.1.3. The Complexity of Tumor Metabolism

As indicated above, the preference by cancer cells for a specific metabolic phenotype is totally related with the three main features of cancer metabolism mentioned before: (a) Heterogeneity, (b) flexibility and (c) adaptability during cancer progression [44].

Metabolic heterogeneity can be a result of the cancer subtype and even different intratumoral cell populations [20,22,44]. Metabolic phenotypes in cancer are both heterogeneous and flexible, and can be the result of combined factors, some intrinsic to the cancer cells and some due to the microenvironment (extrinsic factors). The former include characteristics of the parental tissue and aberrant signaling and gene expression intrinsic to the cells. The extrinsic factors encompass features such as the nutrient milieu, and interactions with the extracellular matrix and stromal cells. In addition, metabolic conditions of the patient (e.g., genetics, obesity, diet) are considered as extrinsic factors influencing cancer metabolism [44,45].

Importantly, metabolic adaptations evolve during cancer progression. The mechanisms involve both well-known and unknown mutations. Briefly, tumor heterogeneity along cancer progression is influenced by several factors: Truncal mutations in tumorigenesis, the accumulation of subsequent mutations, the different combinations of them and the order of arising. An explanation of this scenario is perfectly described by Faubert et al. [44].

### 2.2. Metabolic Phenotype of Pancreatic Cancer

Metabolism is extensively reprogrammed in PDAC as in all other cancers, supporting carcinogenesis, tumor growth and therapy resistance [1,2]. A wealth of investigations has accumulated over the last decade, leading to a better understanding of PDAC metabolism [5,46,47]. These explorations highlight the large degree of PDAC metabolic flexibility [4,48]. Basic research in this field is active to identify new metabolic dependencies that represent new vulnerabilities that can be targeted in the clinic [3,4,5].

Metabolic reprogramming in PDAC shows similarity with other cancers, but also some specificity proper to pancreas and PDAC tumors, such as the presence of a huge stroma (desmoplasia) and poor vascularity leading to nutrient deprivation and hypoxia. Interestingly, PDAC cells are very well adapted to this hostile environment and they can obtain nutrients through both intrinsic adaptation by autophagy [49] and from the microenvironment, i.e., extracellular matrix and neighbor cells [50,51,52,53,54]. By instance, Sousa et al. [52] demonstrated that stroma-associated pancreatic stellate cells support PDAC metabolic needs by alanine secretion, this mechanism being dependent on autophagy. Moreover, Olivares et al. [54] showed that collagen-derived proline is a critical nutrient source in PDAC when other fuels are scarce. Finally, Davidson et al. [53] provided strong evidence that the catabolism of extracellular proteins is necessary in PDAC, showing this statement in tumors from a mutant KRAS-driven mouse model of PDAC.

Another feature of PDAC is common genetic modifications driving cancer development and growth, with almost ubiquitous activating mutations of the KRAS oncogene and frequent inactivating mutations of TP53 and CDKN2A tumor suppressor genes. PDAC arises from cancer precursor lesions known as Pancreatic Intraepithelial Neoplasia (PanIN) and oncogenic KRAS mutations are an early event in low grade PanIN (Figure 2). Further mutations in tumor suppressors are present in high grade PanIN lesion, contributing to disease progression [46]. PDAC can harbor more than 60 mutations, leading to alterations in canonical signaling pathways [55].

The implication of KRAS and p53 proteins in metabolism has been uncovered during the last decade [56,57,58,59]. Activating mutations of KRAS appear early in pancreatic cancer development and were shown to be induced by high glucose concentration, this discovery being instrumental in the better understanding of why diabetes support PDAC initiation [60]. Another metabolic disease, obesity, is also a risk factor of PDAC [61]. Dysregulation of lipid metabolism is one mechanism underlying obesity-driven cancer development [62].

Furthermore, mutations of KRAS in PDAC support dysregulations in several metabolic pathways, such as glucose [24], amino acids [26], fatty acids [63] and nucleotides [64]. In addition, KRAS mutations play a role in the crosstalk with stromal cells [58,65]. It has to be noted here that the metabolic alterations supported by KRAS mutations depend on the tissue context, for example KRAS-driven lung cancer displayed increased BCAA metabolism, being the opposite in PDAC [66]. Finally, KRAS mutations were shown to be associated with epigenetic and gene expression modifications which promote tumorigenesis [67,68].

Moreover, pancreatic cancer cells use a non-canonical pathway for glutamine metabolism: The transamination [26]. Whereas most cancer cells use glutamate dehydrogenase (GLUD1) to convert glutamine-derived glutamate into α-ketoglutarate to refill mitochondrial carbon pool, PDAC use glutamine-derived aspartate to maintain the cellular redox state (Figure 1). For this, PDAC relies on the transaminase enzymes Glutamate Oxaloacetate Transaminase 1 and 2 (GOT1 and GOT2; cytosolic and mitochondrial isoforms, respectively) [26,69]. Importantly, the glutamine metabolic reprogramming in PDAC is regulated by KRAS [26].

Metabolic reprogramming in PDAC is heterogeneous as in other cancers, thus distinct metabolic dependencies can be identified and targeted. For example, Daemen et al. [70] performed a large metabolic profiling in PDAC recognizing different metabolic subtypes associated with glycolytic, lipogenesis and redox pathways. In this study, authors found a “slow proliferating subtype” with low amino-acids and carbohydrates metabolites. More interestingly, they observed two other subtypes with unique metabolic profiles. The “glycolytic subtype” showed high glycolytic and serine metabolites and low levels of metabolites related to redox balance. The other subtype (the most frequent indeed), the “lipogenic subtype”, was enriched in various lipids metabolites such as palmitic acid, as well as OXPHOS metabolites. More importantly, these profiles correlated with sensitivity to different metabolic inhibitors targeting glycolysis, glutaminolysis and lipogenesis.

The mitochondrial compartment remains poorly explored in PDAC. The Warburg effect, postulating a deficiency of mitochondrial respiration, delayed the consideration of mitochondria as actors in cancer metabolism, including PDAC. It is now known that mitochondrial respiration can be active even at low oxygen concentrations (hypoxia) and that fatty acids and amino acids can fuel the TCA in addition to glucose. Recently, our team has demonstrated that mitochondria are still active in PDAC cells and that ATP is produced by OXPHOS along with glycolysis (Masoud et al. paper under consideration). In the same line, Kovalenko et al. [71] identified novel regulators of OXPHOS in PDAC, confirming the notion that mitochondrial metabolism sustains pancreatic cancer. Furthermore, Viale et al. [72] identified a subpopulation of residual pancreatic cancer cells (resistant to KRAS ablation) that showed strong reliance on mitochondrial respiration, and Sancho et al. [73] showed that pancreatic-tumor-initiating cells also depend on mitochondrial OXPHOS. Deciphering further the function and dynamics of the mitochondrial compartment in PDAC is required to unveil new metabolic vulnerabilities and advance in novel therapeutic avenues.

## 3. Mitochondrial Metabolism in PDAC: From Better Knowledge to New Targeting Strategies

### 3.1. Mitochondria Are Hubs in Metabolism

Mitochondrial respiration, through oxidative phosphorylation (OXPHOS), is the primary source of energy in all tissues under aerobic conditions. Despite of being crucial bioenergetics factories, mitochondria are also implicated in several cellular functions including biosynthetic performance, cell signaling and regulation of cell cycle, redox status and programmed cell death. They are involved in physiopathological processes such as aging, neurodegeneration and cancer [74,75]. More importantly, at the center of these processes is mitochondrial metabolism and the efficiency with which it works.

Mitochondria are astounding cytoplasmic organelles with a monophyletic origin from a α-proteobacterial ancestor, which genes were lost or transferred to the eukaryotic (nuclear) genome over millions of years. As consequence, the modern mitochondrial genome resides in two compartments, the mitochondrion (mtDNA) and the nucleus (nDNA), comprising thousands of copies of genes [76]. The mtDNA is replicated independently of the host genome and in humans encodes 13 proteins that are core constituents of OXPHOS. The proteins encoded by the nDNA are imported into the mitochondria and include proteins related also to OXPHOS genes, as well as genes for metabolism and biogenesis [77,78].

Mitochondrial respiration, through OXPHOS, produces approximately 90% of the cellular energy in the form of ATP (Figure 1). To reach this goal, the TriCarboxylic Acid (TCA) cycle uses substrates from glycolysis, Fatty Acid Oxidation (FAO) and amino acid catabolism to generate high-energy electrons (NADH and FADH_2_). These electrons power the electron transport chain (ETC) complexes in the inner membrane, creating a proton force used for high-efficiency ATP generation [79]. This system is embedded in the mitochondrial inner membrane and consists of four multi-subunit complexes: Complex I (CI or NADH: Ubiquinone oxidoreductase), Complex II (CII or succinate: Ubiquinone oxidoreductase), Complex III (CIII or ubiquinol: Cytochrome-c oxidoreductase) and complex IV (CIV or cytochrome-c oxidase). Together with complex V (CV or ATP-synthase), they form the OXPHOS system [80].

Besides being bioenergetics hubs, mitochondria generate building blocks such as amino acids, lipids and nucleotides, as well as they contribute to cytosolic biosynthetic precursors such as acetyl-CoA. Furthermore, mitochondria regulate vital parameters including cytosolic calcium (Ca^2+^) levels, oxidation-reduction (redox) status and generation of most of the Reactive Oxygen Species (ROS). The control of programmed cell death via intrinsic apoptosis is another critical function of these organelles [77].

Importantly, mitochondrial functions are tightly connected to mitochondrial structure and dynamics. Mitochondria are dynamic tubular organelles composed of an outer and inner mitochondrial membrane (OMM and IMM, respectively) that delimitate an intermembrane space (IMS) and the mitochondrial matrix inside the organelle (Figure 1). Mitochondria continually undergo remodeling by fusion and fission events, which influences some of the most important cellular activities [75].

Collectively, this multifunctional nature of mitochondria place them as an integral part of the mechanisms that control cell functioning and survival under physiological and pathological conditions.

### 3.2. Mitochondrial Metabolism and Cancer

The relevance of mitochondrial functions in cancer has been extensively investigated and a huge body of evidence supports the fact that mitochondria are essential for cell survival in different cancer subtypes, including pancreatic cancer [72,81,82]. In accordance, Dong et al. [83] demonstrated that melanoma cells lacking mtDNA exhibited impairment of tumor growth in vivo, which was restored after acquiring mtDNA by transfer of whole mitochondria from the host. Of great interest is that severe suppression of either mitochondrial CI or CII-dependent respiration (by knockdown of NDUFV1 and SDHC subunits, belonging to CI and CII, respectively) resulted in a marked decrease in the ability to form tumors, thus directly associating mitochondrial respiration and cancer growth. Interestingly, Dong et al. [83] point out a major relevance of CI-dependent mitochondrial respiration in cancer formation.

Mitochondrial DNA is more susceptible to damage than nuclear DNA, with a 10–20 fold higher rate of mutagenesis than the nuclear genome. Further, an accumulation of mtDNA mutations has been associated with aging and age-related diseases such as several cancers [78]. In this context, mtDNA mutations and/or reductions in mtDNA copy number that alter the OXPHOS physiology are common features of cancer. This implies that alterations in mitochondrial bioenergetics and metabolism have a role in initiating and/or sustaining the transformed state of cancer [77]. More importantly, genetic and metabolic mitochondrial modifications are implicated not only in metabolic rewiring but also in resistance to therapies, and cancer cells lacking mtDNA or with low copy numbers are much more sensitive to cytotoxic drugs [84,85].

A rational thought is that if mitochondria perform vital functions in normal cells, they will exert these activities in their malignant counterparts. In this line of reasoning, besides energy production, mitochondria confer a biosynthetic advantage to tumor cells, reflected by the production of building blocks (amino acids, lipids and nucleotides) for high-proliferative tumors [86]. By instance, mitochondrial metabolism is highly sustained by two main nutrients, glutamine and fatty acids, that provide electrons to the ETC through NADH production (Figure 1). In addition, glutamine contributes to macromolecular synthesis in ways other than the production of NADPH. Studies in a glioblastoma cell line have shown that glutamine contributes the majority of the cellular oxaloacetate pool, which in turn refill the mitochondrial carbon pool (anaplerosis), with precursors for the maintenance of mitochondrial membrane potential and macromolecules synthesis [28].

Moreover, mitochondria have a key role in apoptosis in mammalian cells. Cancer cells lacking mtDNA show higher apoptosis rates under chemotherapy, leading to enhanced sensitivity to chemotherapeutic drugs [84]. In particular, resistance to Gemcitabine in the pancreatic cancer cell BxPC-3 has been shown to be dependent on mitochondria-mediated apoptosis through the ERK1/2-Bcl-2/Bax signaling pathway [87]. The voltage-dependent anion channel 1 (VDAC1) plays a key role in apoptosis by participating in the release of apoptotic factors from mitochondria to the cytosol. VDAC1 regulates general mitochondrial activity by transporting metabolites, ions, nucleotides and calcium, thus participating in the crosstalk between mitochondria and the cytosol. Its genetic downregulation was shown to induce metabolic reprogramming and tumor growth decrease, suggesting that it could be a good target in cancer therapy [88,89]. Importantly, OXPHOS is the major source of ROS in cells, the mitochondrial complexes I and III being the most notable sites of ROS production. An increase in ROS generation alters mitochondrial membrane potential, inducing damage in the ETC and leading to apoptosis [86]. However, cancer cells are able to keep balanced ROS levels by the production of antioxidants within a window that stimulates proliferation without causing cytotoxicity. Accordingly, Dijk et al. [90] demonstrated that inhibition of mitochondrial protein synthesis combined with Gemcitabine decreases pancreatic cancer cell survival by two means: Reducing cell proliferation (by ATP depletion) and enhancing apoptosis by Gemcitabine (decreasing the mitochondrial inner membrane potential and increasing ROS production).

Altogether, there is compelling evidence that recognizes the fundamental role of mitochondria in all facets of cancer progression, as well as in resistance to chemotherapy. Of more relevance for this review, mitochondrial metabolism is central in the regulation of cancer cell proliferation by bioenergetics and biosynthetic activities.

### 3.3. Targeting Mitochondrial Metabolism in PDAC

Mitochondria are hubs for metabolic processes and trigger metabolic reprogramming, thus sustaining cancer growth and progression. In consequence, mitochondrial energetic metabolism is a very attractive target in cancer therapy, including PDAC. In particular, mitochondrial oxidative phosphorylation (OXPHOS) is an important target in cancer treatment. Other therapeutic strategies include the targeting of glutamine and fatty acids metabolism, as well as the inhibition of TCA cycle intermediates. Figure 3 summarizes the most common approaches to target mitochondrial metabolism in cancer, with a focus on PDAC. Table 1 shows the ongoing or completed clinical trials (clinicaltrials.gov) using mitochondrial metabolism inhibitors specifically to treat PDAC.

#### 3.3.1. Targeting the “OXPHOS Addiction”: Inhibitors of Mitochondrial Respiratory Complexes


Complex I inhibitors


Among the mitochondrial respiratory complexes, Complex I is by far the most frequently described in pancreatic cancer, as well as in other cancers. This interest arose from the observation that the biguanide Metformin -a Complex I inhibitor- improved the outcome in diabetic patients suffering from PDAC [93,94,95,96]. Interestingly, in the retrospective study of Sadeghi et al. [94], this beneficial impact was statistically significant only in patients with non-metastatic disease. However, the combination of Metformin with chemotherapy in clinical trials did not improve survival in patients with PDAC [91,97].

The primary mechanism of action of the biguanides Metformin and Phenformin to impair tumor growth can be attributed to mitochondrial Complex I inhibition, this occurring in pancreatic cancer as well as in other cancers [98,99,100,101,102,103]. Furthermore, several studies showed that biguanides induce a decrease in oxygen consumption and ATP production by mitochondria, promoting glycolysis to compensate inefficient mitochondrial metabolism [100,104,105]. In accordance, Andrzejewski et al. [100] demonstrated that in isolated mitochondria or intact cells, Metformin induces reduced glucose metabolism by the TCA cycle, and that culture in low glucose conditions results in greater sensitivity to Metformin. Altogether, these studies accumulate evidence that Metformin impacts directly mitochondrial metabolism, and that cancer cells counteract this effect by enhanced glycolysis, finally leading to higher lactate production. More importantly, the direct effect of Metformin on mitochondrial Complex I was demonstrated in vivo [98].

Nonetheless, the direct effect of Metformin on Complex I is questionable, and it has been proved that it can exert its antitumoral activity by other means [106,107]. This has led to consider Metformin as a non-specific OXPHOS inhibitor [2]. Some studies described the Metformin anticancer effect through activation of the conserved energy sensor AMP-activated protein kinase (AMPK), which in turn inhibits mammalian target of rapamycin complex 1 (mTORC1) [108,109]. In addition, Metformin-driven AMPK activation was shown to disrupt crosstalk between insulin/IGF-1 receptor and GPCR signaling in pancreatic cancer cells and inhibits the growth of these cells in xenograft models [110,111,112].

On the contrary, another study found that Metformin anti-proliferative activity was AMPK-independent, but its effect was significantly blunted in mTOR-silenced cells [113]. Interestingly, a recent work by Wang et al. [114] revealed that in liver, a pharmacological Metformin concentration promoted mitochondrial respiration by increasing mitochondrial fission in a AMPK-dependent manner. In contrast, supra-pharmacological doses of Metformin reduced mitochondrial respiration but decreasing adenine nucleotide levels and not by inhibiting Complex I per se.

Based on the notion that Metformin effect is by activating AMPK, which in turn suppresses mTORC1 (resulting in autophagy induction), Candido et al. [115] combined a suboptimal dose of Metformin with the mTOR inhibitor Rapamycin. This study showed that Metformin lowered the IC50 of Rapamycin in two PDAC cells (MIA PaCa-2 and BxPC-3), but not in the ASPC-1 cell line. Interestingly, treatment with Metformin by itself did not elicit growth inhibitory effects on BxPC-3 or MIA PaCa-2 cells; however, the Metformin concentration was very low (5 µM). Besides mTORC1, the PI3K/PTEN/Akt/mTORC1 and Raf/MEK/ERK pathways are also inhibited when AMPK is activated. Metformin also improved the effect of a PI3K/mTOR inhibitor in both MIA PaCa-2 and BxPC-3.

In another study, Di Magno et al. [116] investigated the therapeutic action of Phenformin in Hedgehog-dependent medulloblastoma tumors. This work revealed that clinical-relevant dose of Phenformin inhibits the glycerophosphate shuttle, resulting in an increase of redox state/NADH content. Further, this work ruled out the involvement of Complex I, AMPK and mTOR in the antitumoral activity of the biguanide.

Another recent strategy to inhibit PDAC tumorigenesis is based on the role of PDAC stroma. Qian et al. [117] and Duan et al. [118] demonstrated that Metformin reduced the desmoplastic stroma, leading to an improved antitumoral effect by chemotherapy.

Regardless of the mechanism of action, Metformin and Phenformin have shown to possess strong antitumoral activity in vitro and in vivo in several cancers [98,104,105,109,116,119,120,121], including PDAC [112,115,122,123]. This anticancer effect is stronger in the case of Phenformin, probably due to its higher hydrophobicity and permeability. To be noted, the selection of biguanides as anticancer agents should be on the right basis of their mechanism of action, but also including important aspects as the cancer subtype, the heterogeneity between cancer cells and sensitivity to biguanides, the concentration of the drug and the preclinical models used.

In general, all recent and not so recent data reveal that biguanides are an excellent candidate to treat pancreatic cancer. Moreover, the current approach is to use a combined treatment usually with chemotherapeutic agents, glycolytic inhibitors or other compounds [102,115,117,118,124,125]. Interestingly, Gravel et al. [126] demonstrated that dietary restriction of serine and glycine potentiate the antineoplastic effect of Phenformin in allografts models of colon adenocarcinoma. Other works propose the use of Metformin-analogues with a more potent anticancer effect to exert the Metformin-antitumoral effect with pharmacological concentrations [127]. Finally, several clinical trials have been carried out to test the effect of Metformin on PDAC. Table 1 shows the current or completed clinical trials in PDAC, illustrating that these trials are mainly using Metformin in combination with chemotherapy.


Inhibitors of respiratory complexes besides CI


Atovaquone is a FDA approved compound used in the clinic for pneumocystis pneumonia and malaria. Atovaquone is a hydroxy-1, 4-naphthoquinone analogue of ubiquinone, also known as Co-enzyme Q10 (CoQ10) or mitochondrial Complex III. Hence, it has been shown that its anticancer activity is mediated by the inhibition of the mitochondrial CIII, directly decreasing OCR and alleviating tumor hypoxia [128,129].

The Arsenic Trioxide (ATO) is a drug approved by the FDA for the treatment of refractory acute promyelocytic leukemia and it has been investigated in other types of cancer [20]. ATO interferes with OXPHOS by inhibiting mitochondrial Complex IV; in particular, this drug reduces hypoxia leading to an improvement in radiotherapy efficiency [129,130]. In a phase II clinical trial, the effectiveness of ATO was tested in patients who have locally advanced or metastatic pancreatic cancer that has not responded to Gemcitabine (NCT00053222).

#### 3.3.2. Targeting Glutamine Metabolism

Since glutamine catabolism has been widely shown to support pancreatic cancer, several studies have put their effort to target this essential pathway. This targeting was done by different approaches, mainly through the suppression of glutamine uptake by mitochondria or by suppressing glutamine anaplerosis in the cancer cells.

First, to suppress glutamine uptake by mitochondria, the glutamine transporter Glutaminase 1 (GLS1) has been proposed as a target for anticancer treatment [75]. Two distinct chemical compounds inhibiting GLS1 had a growth-suppressive impact on both human and murine PDAC cells [26]. In the same line, Chakrabarti et al. [131] demonstrated that combining GLS1 inhibition (using BPTES or CB-839) with a NADPH:quinone oxidoreductase (NQO1)-bioactivatable drug (β-lapachone) sensitizes PDAC cells to cell death, triggered by ROS burst. Importantly, the authors showed that in order for this combination to be efficient, cells must present both KRAS-driven glutamine dependence and NQO1 expression. Finally, the therapeutic strategy using CB-839 plus β-lapachone was tested in a PDAC preclinical model and was efficient to reduce tumoral growth (compared with either agent alone).

In another study, Biancur et al. [132] addressed the question whether GLS inhibition is an effective therapy in pancreatic cancer. Using the molecule CB-839, authors showed that despite the robust response in vitro, GLS inhibition did not show antitumoral effect in multiple in vivo PDAC models. These findings indicate that pancreatic cancer cells have adaptive metabolic networks in vivo. Through proteomics and metabolomics analyses, the authors identified multiple compensatory pathways mainly related with oxidative stress response, amino acid metabolism, lysosomal processes, glycolysis and pyruvate metabolism. Finally, targeting glutamine metabolism with these adaptive responses may yield clinical benefits.

The second approach to target glutaminolysis is by suppressing glutamine-dependent anaplerosis and is related to the specific route of PDAC to metabolize glutamine: The transamination. In contrast to other cancers, PDAC growth relies on the cytosolic and mitochondrial enzymes GOT1 and GOT2, respectively, and knockdown of related-component enzymes markedly suppressed PDAC growth in vitro and in vivo [26]. Moreover, the use of aminooxyacetate (AOA), a pan-inhibitor of transaminases, robustly inhibited PDAC growing in vitro. Importantly, since this transaminase-mediated pathway is required to sustain PDAC growth, probably through maintaining redox balance, Son et al. [26] suggested to combine with therapies that increase ROS like chemotherapy and radiation.

Interestingly, Yang et al. [69] confirmed the profound impact of the mitochondrial isoform GOT2 in PDAC. To be precise, in this work, the authors showed that GOT2 inhibition induces a significant PDAC senescence by increasing ROS through the cyclin-dependent kinase inhibitor p27. Remarkably, this impact was observed in the cancer cells, but not in non-transformed cells, indicating a potential therapeutic approach for PDAC.

#### 3.3.3. Targeting the Fatty Acid Oxidation

The Fatty Acid Oxidation (FAO) is an outstanding pathway of energy production carried out within the mitochondria. Moreover, FAO is also extremely relevant as a source of NADPH [37]. However, in sharp contrast to the well-studied glycolysis, glutaminolysis and lipogenesis pathways, FAO in cancer has not been well defined. Nonetheless, there is increasing proof that fatty acid catabolism is involved in several aspects of cancer [36].

Different cancer subsets rely on FAO for proliferation, survival, stemness, drug resistance or metastasis [38,39,105,133,134,135,136]. The key enzymes or regulators of FAO have therefore emerged as promising targets for cancer therapy [36,137]. In particular, most of the attention is focused on the rate-limiting enzyme of FAO, the carnitine palmitoyltransferase 1 (CPT1). CPT1 can be pharmacologically targeted by drugs like Etomoxir and Perhexiline or the novel compound Avocatin B, with promising results in vitro and in preclinical studies [135,138,139,140,141,142].

Notwithstanding, the role of FAO in pancreatic cancer has not yet been elucidated and remains as an unexplored area of cancer metabolism. This fact is quite surprising due to the presence of a lipid-rich environment around the pancreas, which has been shown to be important in other cancers [143,144]. To our knowledge, only Shin et al. [145] have reported that FAO supports cell viability and invasion of PDAC in vitro under acidic extracellular conditions.

#### 3.3.4. Targeting the TCA Cycle: Inhibitors of Metabolic Intermediates

The TCA cycle (also known as Krebs cycle) provides the electron donors NADH and FADH_2_ which fuel the ETC to drive electrochemical proton gradient for ATP production. Moreover, the activity of this cycle is not restricted to energy performance, but its relevance lies on the fact that major metabolic pathways converge on it: Glycolysis, glutaminolysis and FAO. These converging pathways fill up the TCA cycle with key metabolic intermediates that serve as substrates for anabolism processes, such as citrate, isocitrate, α-ketoglutarate, succinate, fumarate, malate and oxaloacetate.

The importance of TCA cycle intermediates in cancer is widely recognized. By instance, mutations in isocitrate dehydrogenase enzymes 1 and 2 (IDH1 and IDH2) play a critical role in tumorigenesis. Hence, several IDH inhibitors are considered as anticancer agents [146]. IDH enzymes catalyze the conversion of isocitrate to α-ketoglutarate and when mutated, generate the oncometabolite D-2-hydroxyglutarate (D-2HG) which inhibits epigenetic enzymes. Mutations of these enzymes have been found in cancers such as glioblastoma [79,147]. However, even if IDH1 is a commonly mutated metabolic enzyme in some human cancers, only a recent report did describe IDH1 mutation in one patient with PDAC [148].

A more relevant TCA cycle-target in PDAC is currently under clinical trials using an inhibitor of α-ketoglutarate dehydrogenase and pyruvate dehydrogenase enzymes [9]. Devimistat (also known as CPI-613) is a selective-cancer agent that is currently on phase II of clinical trials for patients with unresectable pancreatic cancer (NCT03699319). The objective of this assay is to treat patients with Devimistat in combination with modified FOLFIRINOX (mFOLFIRINOX). Concerning patients with metastatic PDAC, a phase III study to evaluate mFOLFIRINOX plus Devimistat is also ongoing (NCT03504423). In a small cohort phase I-study [92], patients with metastatic PDAC showed 61% of response rate to Devimistat combined with mFOLFIRINOX. In addition, the combination regimen was safe and well tolerated, encouraging further exploring this therapeutic strategy (NTC01835041).

A number of compounds already exist in the cancer field to target the TCA cycle intermediates as well as the reaction converting pyruvate to acetyl-CoA for entering the cycle. However, most of these compounds are poorly explored in PDAC, which opens the possibility to test their effectiveness in this pathology.

## 4. Conclusions

The last decade has witnessed an explosion of research on the metabolism of PDAC. Metabolic specificities and dependencies of pancreatic tumors are regularly uncovered. This research is very promising to identify vulnerabilities that can be targeted in the clinic. In that context, energetic metabolism still requires more attention on PDAC, considering mitochondrial respiration and not only glycolysis. Mitochondria appear now to be central in cancer cell survival and resistance to therapies. Combining Metformin (inhibiting the respiratory Complex I) with chemotherapy proved to be disappointing, but a way to identify the patients likely to respond to this combination is still lacking. By contrast, current clinical trials targeting the TCA cycle with Devimistat in combination with chemotherapy are very encouraging. Investigation of mitochondrial vulnerabilities in PDAC, which is still in its infancy, deserves further consideration in basic, translational and clinical research.

## Figures and Tables

**Figure 1 biomedicines-08-00270-f001:**
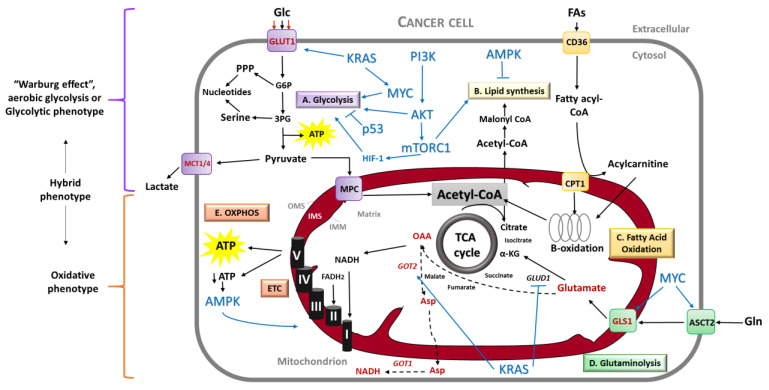
Metabolic reprogramming in cancer. In contrast with normal cells, cancer cells are able to fulfill their metabolic needs with minimal dependence on extrinsic stimuli by growth factors, through deregulation of pathways related to energy production and biosynthesis. Glycolysis and glutaminolysis are by far the most investigated pathways. The most common signaling molecules involved in metabolic pathways are PI3K, AKT, mTORC1, AMPK, as well as the oncogenic proteins KRAS and MYC, and the tumor suppressor p53 (depicted in blue font). **A**. Glycolysis. The “Warburg effect” or aerobic glycolysis is characterized by high glycolytic rates observed in cancer cells even in the presence of oxygen, with a consequent high amount of lactate production. KRAS induces glucose (Glc) uptake by cancer cells through upregulation of the GLUT1 transporter. The glycolysis pathway is promoted by KRAS, MYC, AKT, loss of the tumor suppressor TP53 and HIF-1. Additionally, glycolytic intermediates are diverted towards biosynthetic processes, mainly DNA and RNA synthesis by the Pentose Phosphate Pathway (PPP). Glycolysis allows the production of pyruvate and ATP. Pyruvate can be converted into lactate and shuttled outside the cell; PDAC cells have enhanced activity of MCT1/4. Pyruvate can also enter the mitochondria through the MPC for further feeding the TCA cycle. **B**. Lipid synthesis. Cancer cells activate lipid synthesis (also named fatty acid synthesis or lipogenesis) to meet their demands. This pathway requires the production of cytosolic Acetyl-CoA, which is mainly derived from mitochondrial citrate into the TCA cycle. **C**. Fatty Acid Oxidation (FAO). FAO comprises a cyclical series of reactions that result in the shortening of fatty acids (FAs) molecules (beta-oxidation) to produce Acetyl-CoA and NADH/FADH_2_. CPT1, the rate-limiting enzyme of FAO, conjugates fatty acids with carnitine to translocate them into the mitochondria, where the acylcarnitines undergo B-oxidation. **D**. Glutaminolysis. In PDAC, KRAS and MYC also rewires glutamine (Gln) metabolism. Gln is transported into mitochondria by GLS1, then converted into glutamate and finally aspartate (Asp), which is shuttled to the cytosol to generate NADH for redox balance. PDAC relies on the transaminase enzymes GOT1 and GOT2 for this glutamine-metabolic rewiring. **E**. OXPHOS. Mitochondria are functional in cancer, and several cancer subtypes or populations exhibit an oxidative phenotype. Mitochondrial respiration, through OXPHOS, produces most of the cellular ATP. For this, the TCA cycle oxidizes substrates from glycolysis, FAO and glutaminolysis to produce high-energy electron donors (NADH and FADH_2_). These electrons power the ETC complexes in the mitochondrial inner membrane, creating a proton force used for high-efficiency ATP generation. Upon detection of decreased energy charge, AMPK increases energy generation by enhancing OXPHOS. Metabolites and enzymes overexpressed in PDAC are depicted in red font. Dashed arrows represent serial reactions. α-KG, α-ketoglutarate; Akt, protein kinase B; AMPK, adenosine monophosphate-activated protein kinase; 3PG, 3-phosphoglycerate; ASCT2, alanine/serine/cysteine transporter 2; ATP, adenosine 5′-triphosphate; CPT1, carnitine palmitoyltransferase 1; ETC, electron transport chain; FADH_2_, flavin adenine dinucleotide; G6P, glucose-6-phospate; GLS1, glutaminase 1; GLUD1, glutamate dehydrogenase 1; GLUT1, glucose transporter 1; GOT1 and 2, glutamate oxaloacetic transaminase 1 and 2; HIF-1, hypoxia-inducible factor 1; IMM, inner mitochondrial membrane; IMS, intermembrane space; MCT1/4, monocarboxylate transporter 1 and 4; MPC, mitochondrial pyruvate carrier; mTORC1, mammalian target of rapamycin complex 1; NADH, nicotinamide adenine dinucleotide; OAA, oxaloacetate; OMS, outer mitochondrial membrane; OXPHOS, oxidative phosphorylation; PI3K, phosphatidylinositol 3-kinase; PPP, pentose phosphate pathway; TCA, tricarboxylic acid.

**Figure 2 biomedicines-08-00270-f002:**
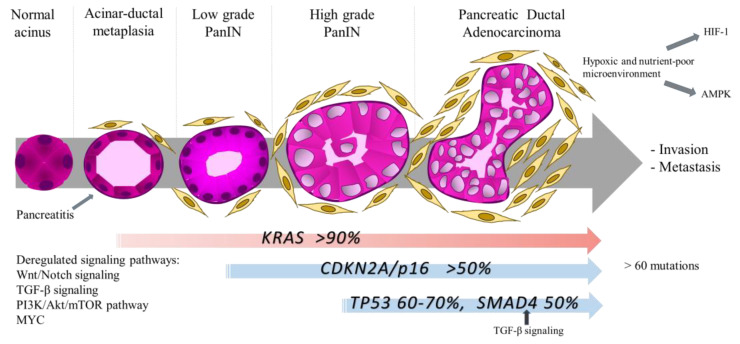
Schematic representation of genetic and signaling pathways alterations in PDAC carcinogenesis. PDAC arises from precursor lesions which are acinar-ductal metaplasia and Pancreatic Intraepithelial Neoplasia (PanIN). KRAS mutations are an early event in low grade PanIN and further mutations in tumor suppressor genes as CDKN2A/p16, TP53 and SMAD4 are present in high grade PanIN lesions, contributing to disease progression. Mutations in these four genes are by far the most common in PDAC; however, PDAC can harbor more than 60 mutations. Besides genetic alterations, deregulated signaling pathways and stromal-associated factors promote an aggressive PDAC behavior. Akt, protein kinase B; AMPK, adenosine monophosphate-activated protein kinase; CDKN2A/p16, cyclin dependent kinase inhibitor 2A; HIF-1, hypoxia-inducible factor; mTOR, mammalian target of rapamycin; PI3K, phosphatidylinositol 3-kinase; SMAD4, Mothers against decapentaplegic homolog 4; TGF-β signaling, transforming growth factor-β signaling.

**Figure 3 biomedicines-08-00270-f003:**
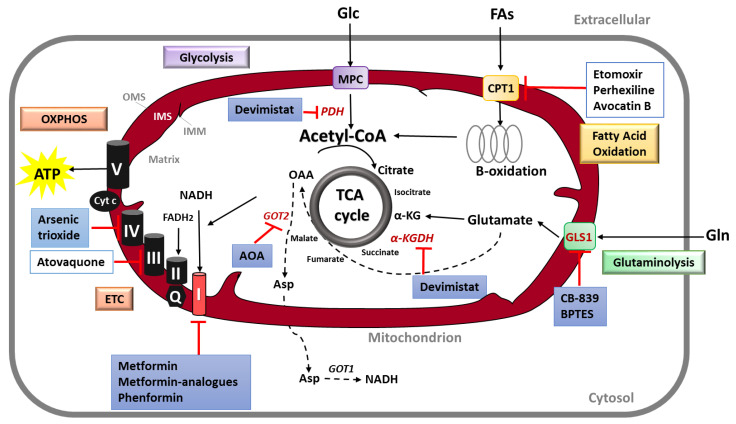
Targeting mitochondrial metabolism in PDAC. Several strategies have been developed to target mitochondrial metabolism in cancer. Compounds that target the ETC and OXPHOS function are an important target in cancer therapy. Among these, molecules that inhibit the mitochondrial Complex I such as the biguanides Metformin and Phenformin are the most frequent described. Other OXPHOS inhibitors are Arsenic trioxide and Atovaquone, targeting CIV and CIII, respectively. Other therapeutic strategies include the targeting of glutamine and fatty metabolism uptake, by inhibiting the enzymes that allow the entry of the molecules to mitochondria (GLS1 and CPT1 for glutamine and fatty acids, respectively). Another way to target mitochondrial metabolism is by suppressing a specific route of PDAC for metabolize glutamine: The transamination. The use of aminooxyacetate (AOA), a pan-inhibitor of transaminases, is able to inhibit the mitochondrial enzyme GOT2. Finally, the inhibition of TCA cycle intermediates is another strategy for cancer therapy. Devimistat (also known as CPI-613) is a selective-cancer agent that inhibits the enzyme α-ketoglutarate dehydrogenase and the pyruvate dehydrogenase upstream to the TCA cycle; this compound is currently under clinical trials for PDAC treatment. The compounds tested for PDAC treatment are in blue boxes. α-KG, α-ketoglutarate; α-KGDH, α-ketoglutarate dehydrogenase; ATP, adenosine 5′-triphosphate; CPT1, carnitine palmitoyltransferase 1; ETC, electron transport chain; FADH_2_, flavin adenine dinucleotide; GLS1, glutaminase 1; GOT1 and 2, glutamate oxaloacetate transaminase 1 and 2; IMM, inner mitochondrial membrane; IMS, intermembrane space; MPC, mitochondrial pyruvate carrier; NADH, nicotinamide adenine dinucleotide; OAA, oxaloacetate; OMS, outer mitochondrial membrane; OXPHOS, oxidative phosphorylation; PDH, pyruvate dehydrogenase; TCA, tricarboxylic acid.

**Table 1 biomedicines-08-00270-t001:** Current and completed clinical trials using mitochondrial metabolism inhibitors in PDAC.

Targeted Function	Molecular Target	Mitochondrial Inhibitor	Additional Treatment	PDAC Stage	Clinical Trial Phase	NCT Number
**OXPHOS**	Complex I	Metformin	Gemcitabine, Erlotinib	Locally advanced or metastatic	II	NCT01210911, ref. [91]
			Gemcitabine, Nab-paclitaxel, dietary supplement	Unresectable	I	NCT02336087
			Oxaliplatin, Leucovorin calcium, Fluorouracil	Metastatic	II	NCT01666730
			Stereotactic radiosurgery	Borderline-resectable or locally advanced	Early phase I	NCT02153450
			Paclitaxel	Locally advanced or metastatic, after Gemcitabine failure	II	NCT01971034
			Rapamycin	Metastatic, stable disease after FOLFIRINOX or Gemcitabine treatment	I	NCT02048384
	Complex IV	Arsenic trioxide	---------	Locally advanced or metastatic, after Gemcitabine failure	II	NCT00053222
**TCA cycle**	PDH and α-KGDH	Devimistat (CPI-613)	mFOLFIRINOX	Unresectable	II	NCT03699319
				Metastatic	III	NCT03504423
					I	NTC01835041, ref. [92]

α-KGDH, α-ketoglutarate dehydrogenase; mFOLFIRINOX, modified FOLFIRINOX; NCT, National Clinical Trial; OXPHOS, oxidative phosphorylation; PDH, pyruvate dehydrogenase; TCA, tricarboxylic acid.

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
