# Peer review of "Mitochondrial Metabolism in PDAC: From Better Knowledge to New Targeting Strategies"

_biomedicines, 2020, doi:10.3390/biomedicines8080270_

Round 1

Reviewer 1 Report

The authors of the paper conducted a synthetic review of the latest literature on the role of mitochondrial metabolism in cancer with a particular focus on pancreatic ductal adenocarcinoma. Moreover, they present the latest therapeutic strategies targeting this metabolism. In my opinion, the work is a valuable review summarizing available and suggesting future therapeutic fields/options in such a difficult to treat disease as pancreatic cancer. However, I have some minor comments about the manuscript. There should be a citation/s at the end of the sentence in the Background chapter, on page 2, line 8. Furthermore, Figure 2 shows that Devimistad inhibits a-ketoglutarate. This is not a precise presentation, because this compound inhibits the enzyme involved in the conversion of AKG to succinyl CoA, rather than AKG alone (the AKG pool may thus increase). Also, the description in the legend of Figure 2: “the enzyme a-ketoglutarate” is incorrect and should be corrected.

Author Response

Reviewer 1

The authors of the paper conducted a synthetic review of the latest literature on the role of mitochondrial metabolism in cancer with a particular focus on pancreatic ductal adenocarcinoma. Moreover, they present the latest therapeutic strategies targeting this metabolism. In my opinion, the work is a valuable review summarizing available and suggesting future therapeutic fields/options in such a difficult to treat disease as pancreatic cancer. However, I have some minor comments about the manuscript.

There should be a citation/s at the end of the sentence in the Background chapter, on page 2, line 8.

Response: We did add five citations at the end of the sentence “Specifically, pancreatic cancer is a disease that could greatly benefit from mitochondrial metabolism targeting” line 39 (now references 1 to 5).

Furthermore, Figure 2 shows that Devimistad inhibits a-ketoglutarate. This is not a precise presentation, because this compound inhibits the enzyme involved in the conversion of AKG to succinyl CoA, rather than AKG alone (the AKG pool may thus increase).

Response: We totally agree with the reviewer and thank him/her for putting the attention on our mistake. We did modify the text (line 574) by adding “dehydrogenase” after “α-ketoglutarate”. We did modify the Figure accordingly (New Figure 3, previously Figure 2). As Devimistat/CPI-613 is inhibiting mitochondrial metabolism also through inhibition of another dehydrogenase enzyme, namely Pyruvate Dehydrogenase, we did also modify the New Figure 3 to add this information.

Also, the description in the legend of Figure 2: “the enzyme a-ketoglutarate” is incorrect and should be corrected.

Response: Related to the previous point, we did modify the new Figure 3 and accordingly the legend of new Figure 3 (lines 1199-1200, 1202, 1208).

Additional modifications:

Figure 1: We depicted the Warburg effect and oxidative phenotype in order to be more in accordance with the text. In the legend, we added sentences in lines 1132-1134 and 1153-1154. Also, in the Figure we did precise that the energy sensor AMPK enhances OXPHOS in case of need (sentence in legend lines 1159-1160). Finally, in the legend of the Figure, we included the meaning of the abbreviations Akt, AMPK, mTOR, and PI3K, which were missing in the first version of the paper (lines 1162-1163, 1169-1172).

Reviewer 2 Report

In this review article Gabriela Reyes-Castellanos et al. summarizes mitochondrial metabolism in PDAC and targeting mitochondrial metabolism is thus an exciting area in the treatment of various types of cancer including PDAC.

In general this review article was well constructed, and extensively documented on mitochondrial tumor metabolism. This article is recommended for publication in Biomedicines, once the following items have been addressed.

Using schematic representation show pathological stages and dysregulated molecular events in PDAC development as figure 1. For example different PDAC development process categorized into pre-cancerous lesion and malignant PDAC according to the pathological grade. Simultaneously, show different aberrant genetic alterations occurrence at different stages of PDAC development.

One of the most prominent characteristics of PDA is an intense desmoplastic reaction around the tumor; it will be interesting to include a section on Metabolic Crosstalk in the PDAC Tumor Microenvironment.

Show in tabulated form all clinical trials of inhibitors targeting metabolism in patients with PDAC for easy understanding. Such as Metabolism-modulating agents in clinical development for pancreatic ductal adenocarcinoma therapy

To adapt severely metabolic constraints, PDAC cells rely on specific metabolic reprogramming, offering potentially innovative strategies to treat patients with PDAC in the future. What are the different Subtypes of PDAC based on metabolic features arrange them in tabulated form.

Cite following article, although these article are not related to PDAC, but they nicely shows targeting specific protein induces metabolic reprogramming in various solid cancer

Arif T et al 2018. Mitochondrial VDAC1 silencing leads to metabolic rewiring and the reprogramming of tumour cells into advanced differentiated states

Arif T et al, 2017. VDAC1 is a molecular target in glioblastoma, with its depletion leading to reprogrammed metabolism and reversed oncogenic properties.

Author Response

Reviewer 2

In this review article Gabriela Reyes-Castellanos et al. summarizes mitochondrial metabolism in PDAC and targeting mitochondrial metabolism is thus an exciting area in the treatment of various types of cancer including PDAC.

In general this review article was well constructed, and extensively documented on mitochondrial tumor metabolism. This article is recommended for publication in Biomedicines, once the following items have been addressed.

Using schematic representation show pathological stages and dysregulated molecular events in PDAC development as figure 1. For example different PDAC development process categorized into pre-cancerous lesion and malignant PDAC according to the pathological grade. Simultaneously, show different aberrant genetic alterations occurrence at different stages of PDAC development.

Response: We thank the reviewer for this suggestion that will improve the visual impact of our review, and more importantly, for illustrating the genetic and signaling pathway alterations during the progression of PDAC. For this, we added a new Figure (Figure 2), that is mentioned in the section 2.2 “Metabolic phenotype of pancreatic cancer” (lines 246-251). In parallel, we added a new reference that is depicted in red color in the references section (reference #55, Iacobuzio-Donahue et al., 2012, citation line 250, and reference in lines 766-769).

One of the most prominent characteristics of PDA is an intense desmoplastic reaction around the tumor; it will be interesting to include a section on Metabolic Crosstalk in the PDAC Tumor Microenvironment.

Response: We thank the reviewer for this suggestion and we totally agree that PDAC microenvironment plays a critical role in all stages of the disease. For this reason, we developed this subject in more detail in the section 2.2 “Metabolic phenotype of pancreatic cancer” (line 237-243). However, we believe that even if it’s a very important subject to pay attention on, it is not the main scope of this article, for that we decided to keep it in the section devoted to PDAC metabolic phenotype.

Show in tabulated form all clinical trials of inhibitors targeting metabolism in patients with PDAC for easy understanding. Such as Metabolism-modulating agents in clinical development for pancreatic ductal adenocarcinoma therapy

Response: We thank the reviewer for this suggestion. As the scope of our review is mitochondrial metabolism in PDAC, we focused our manuscript on the current knowledge and targeting strategies specifically concerning mitochondrial metabolism. However, we thought that it was a very good idea to add a table showing all clinical trials of compounds targeting mitochondrial metabolism specifically in PDAC (new Table 1). We included the clinical trials that are in progress or completed. We mention this table at first in the 3.3 section (lines 408-409). This table illustrate that clinical trials targeting PDAC are mainly using Metformin, which was missing in the first version of the Review. These clinical trials are mentioned in the lines 479-482. Finally, we added a clinical trial using Arsenic Trioxide for treating PDAC (lines 492-495).

To adapt severely metabolic constraints, PDAC cells rely on specific metabolic reprogramming, offering potentially innovative strategies to treat patients with PDAC in the future. What are the different Subtypes of PDAC based on metabolic features arrange them in tabulated form.

Response: We totally agree with Reviewer 2 that targeting metabolic reprogramming is promising for future efficient treatments in PDAC. In the case of mitochondrial metabolism which is the focus of our manuscript, targeting the TCA cycle with Devimistat is already in clinical trials with encouraging data. We totally agree that novel metabolic targeting approaches require the stratification of patients in order to identify the ones most likely to respond to one treatment or another. To our knowledge, the only publication reporting metabolic subtypes in PDAC is the Daemen et al. PNAS 2015 paper (reference #70). In our laboratory, we are currently developing projects aiming at stratifying the PDAC patients according to features in mitochondrial metabolism (Masoud et al., in revision; mentioned line 289). The latter publication will be cited in this review whether it will be accepted at the time of publication.

Cite following article, although these article are not related to PDAC, but they nicely shows targeting specific protein induces metabolic reprogramming in various solid cancer

Arif T et al 2018. Mitochondrial VDAC1 silencing leads to metabolic rewiring and the reprogramming of tumour cells into advanced differentiated states

Arif T et al, 2017. VDAC1 is a molecular target in glioblastoma, with its depletion leading to reprogrammed metabolism and reversed oncogenic properties.

Response: We also found interesting to add these two articles in our review. We added sentences lines 379 to 385.

Additional modifications:

Figure 1: We depicted the Warburg effect and oxidative phenotype in order to be more in accordance with the text. In the legend, we added sentences in lines 1132-1134 and 1153-1154. Also, in the Figure we did precise that the energy sensor AMPK enhances OXPHOS in case of need (sentence in legend lines 1159-1160). Finally, in the legend of the Figure, we included the meaning of the abbreviations Akt, AMPK, mTOR, and PI3K, which were missing in the first version of the paper (lines 1162-1163, 1169-1172).
